# Percolation in networks with local homeostatic plasticity

Giacomo Rapisardi[1,2], Ivan Kryven [3,4] & Alex Arenas [1✉]

Percolation is a process that impairs network connectedness by deactivating links or nodes. This process features a phase transition that resembles paradigmatic critical transitions in epidemic spreading, biological networks, traffic and transportation systems. Some biological systems, such as networks of neural cells, actively respond to percolation-like damage, which enables these structures to maintain their function after degradation and aging. Here we study percolation in networks that actively respond to link damage by adopting a mechanism resembling synaptic scaling in neurons. We explain critical transitions in such active networks and show that these structures are more resilient to damage as they are able to maintain a stronger connectedness and ability to spread information. Moreover, we uncover the role of local rescaling strategies in biological networks and indicate a possibility of designing smart infrastructures with improved robustness to perturbations.

[1] Departament d'Enginyeria Informàtica i Matemàtiques, Universitat Rovira i Virgili, E-43007 Tarragona, Spain. [2] Barcelona Supercomputing Center (BSC), Barcelona, Spain. [3] Mathematical Institute, Utrecht University, Budapestlaan 6, 3508 TA Utrecht, The Netherlands. [4] Centre for Complex Systems Studies, 3584 CE Utrecht, The Netherlands. ✉email: alexandre.arenas@urv.cat

The resilience of complex networks to withstand failures and attacks is one of their most intriguing properties[1]. When edges have equal strengths, resilience can be studied with the mathematical framework of *percolation*[2]—a stochastic process akin to the permeation of a liquid through a porous membrane. This process removes edges uniformly at random with some given probability. The large-scale connectivity of the network is then studied by tracking the size of the largest connected component (LCC) as a function of this probability[3,4]. The size of LCC indicates what fraction of the whole network stays connected after damage, and in contexts such as transportation or communication—connected means functional. Studying percolation helps to understand the contribution of the network structure to its resilience, and both first- and second-order phase transitions in the size of LCC have been observed across a wealth of studies[1,4–8]. The most notable example refers to explaining the spreading of a disease driven by the susceptible–infected–recovered contact process. Percolation in a random network is identical to this process when the infectious time is constant[9] and leads to a useful approximation otherwise[10]. Percolation has been also used to understand the early stages of formation of the brain by probing how resilient are interconnected neuronal cultures in vitro[11,12].

When links feature different strengths (also called weights or capacities), both the structure and link weights contribute to network resilience, for example, as it happens with road and airline networks, the Internet, electricity grid, and financial networks. The role of such weights in network resilience can be mathematically studied with *filtration*[13,14], a process that removes all edges with a weight less than a given threshold. This process can be thought of as a permeation of a liquid with insoluble particles through a porous membrane, wherein the particles may flow through a pore only after they have clogged it up, otherwise, they remain filtrated.

Optimizing the distribution of link weights to secure a more resilient network is an old problem: in epidemiology, it is known as targeted immunization[15], but it also occurs in electrical engineering when designing efficient power grids, and urban planning when improving the traffic flow capacity of road networks. Apart from wide use in the top-down design, several biological complex systems were also observed to employ local self-regulation of link strengths with a positive effect on the global functioning of the whole network[16,17]. Both brain networks and food webs are believed to perform link strengths optimization by virtue of a self-regulatory mechanism that is triggered in response to damage.

In such systems, filtration-like processes are accompanied by an *active response* that mitigates the damage due to removed links. Neurons are hypothesized to control their activity using *synaptic scaling*[18,19], a mechanism that allows neurons to adjust their synaptic strengths to conserve the overall neural activity despite external perturbations or damage[19–21]. In a similar fashion, food webs representing the dynamical system for interspecies mass/energy transporting have also been observed to feature a high degree of coherence that may be a consequence of self-regulation and adaptive rewiring[22–25]. Load redistribution and rewiring mechanisms are thought to have shaped the core-periphery structure of the world airline network[26]. At the same time, self-regulation-inspired principles were also proposed to be used in top-down intervention scenarios for preventing species extinction in ecosystems[27].

## Results

In the remainder, we present a mathematical theory for damage-response processes in complex networks that explains how maintaining local conservation of the total in- (or out-) weights of a node reflects on the global connectedness of the whole system.

To this end, we consider a directed random network model with positive weights on edges. An instance of such a network is defined by a weighted adjacency matrix $A_{ij} \in \mathbb{R}^+$ if node $i$ points to node $j$, otherwise $A_{ij} = 0$.

We model the adaptive network degradation by several iterations of the filtration stage, followed by an update of the weights of the survived edges according to a *homeostatic plasticity* principle, that is conservation of the sum of all *in-weights* for each node. This principle is inspired by the conservation of the total pre-synaptic strength observed in neurons[20]. Using this theory, we show that a simple local self-regulatory mechanism actively adjusting link weights in a fashion similar to the synaptic scaling in neurons[20], may significantly improve large-scale connectivity of the whole network and hence maintain network functioning, even if the damage has caused a loss of a large fraction of links. Note that the concept of homeostatic plasticity we adopt here is different from the one of *homeostasis* that usually appears in the literature of dynamical systems[28,29], which instead is generally related to the properties of stable fixed points in systems of ODEs.

To consider a general setting, our directed network model is defined by an arbitrary degree-weight distribution, $f_k(x)$, the probability that a uniformly chosen directed edge has weight in the interval $[x, x + dx]$ and terminates at the node of in-degree $k$. The joint distribution can be factorized, $f_k(x) = l_k w_k(x)$, where the *excess-degree* distribution $l_k$, $\sum_{k \geq 0} l_k = 1$ is the probability that a randomly chosen edge terminates at a node of in-degree $k$, and $w_k(x)$ satisfying $\int_0^\infty w_k(x)\mathrm{d}x = 1$ is the probability density function for edges that terminate at a node of in-degree $k$. One step of the damage-response cycle is introduced as an operator $\mathcal{A}$ acting on the degree-weight distribution, $f_k^1(x) = \mathcal{A}f_k^0(x)$, where $f_k^0$ is the distribution before the edge removal and $f_k^1$ is the distribution after the damage-response cycle. This operator can be further decoupled as a convolution product of the *damage* $\mathcal{D}_{k,y}$ and *response* $\mathcal{R}_{k,y}$ operators

$$\mathcal{A}f_k(x) = \sum_{n \geq k} \left[ \mathcal{D}_{k,y}(f_n(x)) \right] * \left[ \mathcal{R}_{k,y}(f_n(x)) \right]. \quad (1)$$

Here, the contribution to the probability density of degree-$k$ nodes comes from the nodes with higher degrees due to the fact that some edges are removed but none are added, and the surviving edges increase their weights as captured by the convolution operation ("$*$") along the weight dimension $x$. See the "Methods" section for more details.

Our operator $\mathcal{A}$ provides a general framework for studying the phenomenon of adaptive degradation. Let us now consider a particular instance of degradation/response mechanism wherein the former is represented by *filtration*, removing all edges with the weight below a given threshold $y$, and the latter is given by the following redistribution rule. For a node of degree $k$, the weights of the in edges $x_i$, $i = 1, \dots, k$ are updated according to

$$\begin{cases} x_i \to x_i + \frac{\Delta}{m}, & x_i \geq y, \\ x_i \to 0, & x_i < y, \end{cases} \quad (2)$$

where $m$ is the total number of edges for which $x_i > y$, and $\Delta = \sum_{\{i, x_i < y\}} x_i$ is the sum of all removed weights. A simple check shows that similarly to homeostatic response in neurons, $\sum_i x_i$ is conserved. Methods section provides the explicit forms of $\mathcal{D}_{k,y}$ and $\mathcal{R}_{k,y}$ for this mechanism.

In the first panel of Fig. 1, we present a sketchy representation of the process described by rule (2), while the second panel illustrates multiple successive steps of the filtration process with and without homeostatic response on an example of a small synthetic network. One can see that the homeostatic response mechanism mitigates the degradation process by enabling the network to withstand larger filtration thresholds. In Fig. 2 we use

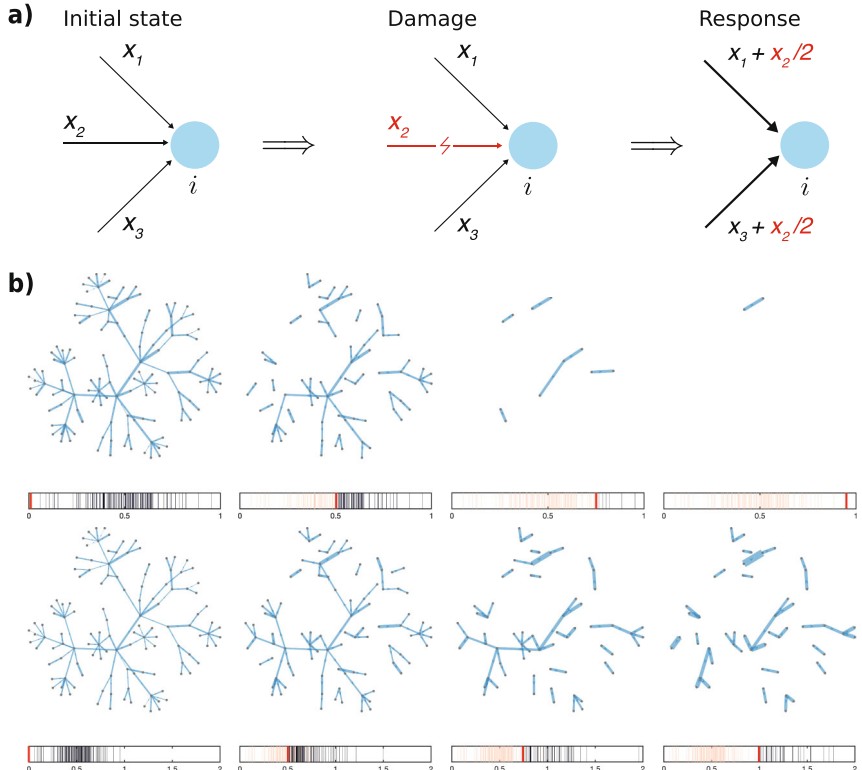

**Fig. 1 Simple filtration vs. filtration with homeostatic response. a** Pictorial representation of the damage–response process for a given node $i$. Here, in the damaged state, one of the three initial in-edges is removed, hence its weight, $w_2$, is equally redistributed among the surviving edges, in order to conserve the local in-strength of the initial state. **b** Several snapshots illustrating an example of multi-step degradation on a small network, governed by: filtration, where every edge is removed if its weight is below a given threshold (first row), and filtration with the homeostatic response that maintains the total weight of in-edges at a constant level (second row). The bars below the panels indicate the distribution of edge weights (*black*) and the value of the threshold *red*.

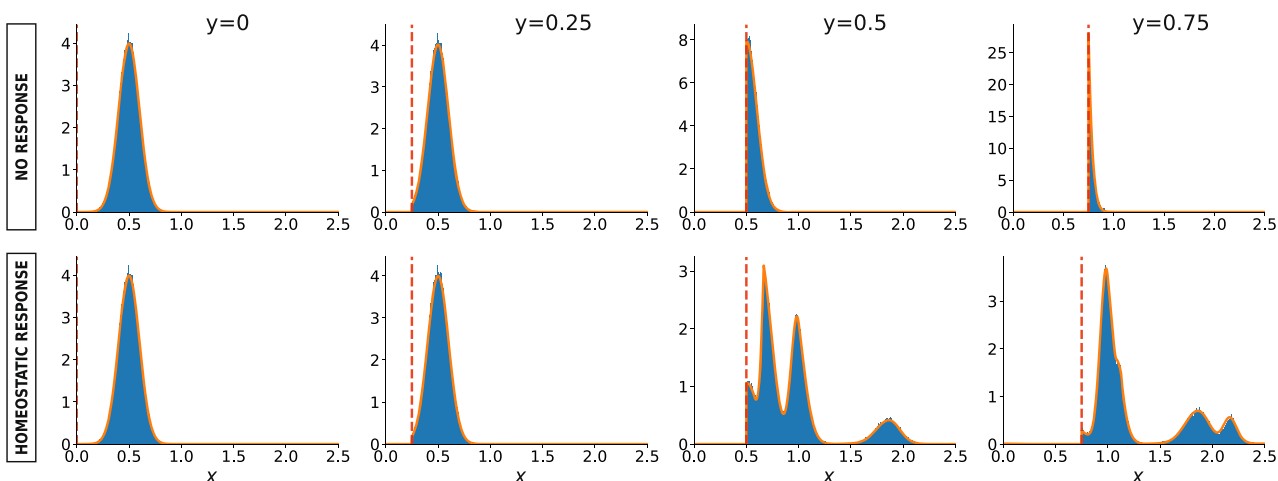

**Fig. 2 Impact of the homeostatic response on the weight distribution.** Evolution of the *total* weight distribution $w(x)$ on four successive instances of filtration without response (upper panel), or with the homeostatic response (lower panel), on a large random regular graph of $N = 5 \times 10^4$ nodes, degree $z = 4$ and normally distributed edge weights, at increasing values for the threshold $y$. The network has been generated according to the directed configuration model[42], while the edge weights are distributed according to Gaussian distribution with mean $\mu = 0.5$ and standard deviation $\sigma = 0.1$. Each plot shows the empirical weight distribution from stochastic simulations (blue histogram) compared with the prediction derived from our model (orange solid line). The vertical dashed red line in each plot represents the value of $y$.

master equation (1) to quantify the evolution of the degree-weight distribution caused by the multi-step damage/response process in a larger synthetic random network of $N = 5 \times 10^4$ nodes, and normally distributed weights. Validation of these predictions with stochastic simulations shows that our equation is highly capable to predict complex patterns revealed by the simulations.

From the structural point of view, the connectivity of the whole network is characterized by the largest component in which

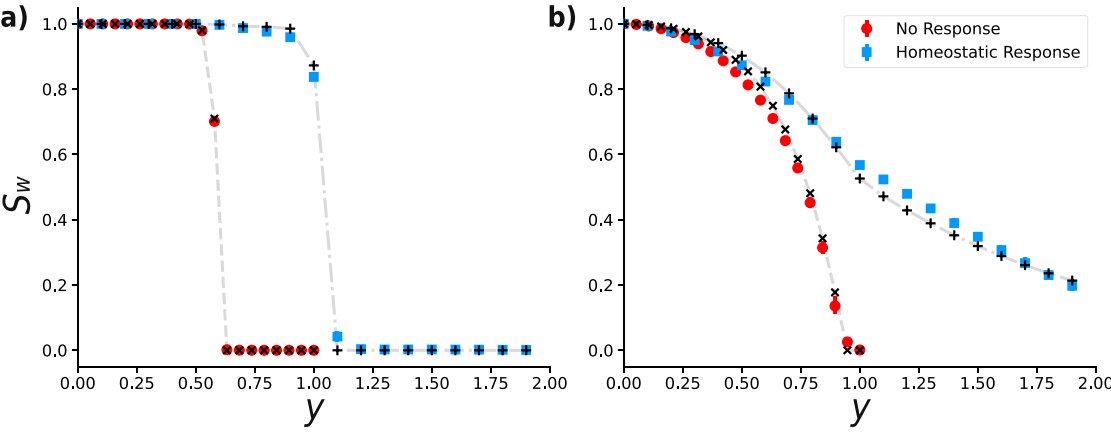

**Fig. 3 Homeostatic response delays the disruption of the giant component.** Evolution of the weak giant component's size, $S_W$, as a function of the threshold $y$, for **a** a regular random graph of $N = 2 \times 10^4$ nodes and degree $z = 4$ with Gaussian distributed weights (the same distribution of Fig. 2), and **b** a brain network of $N = 983$ nodes representing a volume of the mouse neocortex[31, 32] with uniformly distributed weights in the interval [0,1]. Colored markers represent the outcome of Monte Carlo simulations, obtained by averaging over 100 realizations of the weight distribution for the regular graph, and 1000 realizations for the real dataset. In both cases, we considered 20 successive steps of filtration without response (red circles) and with the homeostatic response (blue squares). Black markers represent the corresponding estimates from the model for each value of $y$, while gray dashed and dash-dotted lines represent linear interpolations between theoretical values, which serve only as visual guidelines.

nodes are path-connected regardless of the direction of the links —the largest *weakly* connected component. We estimate the size of this component, $S_W$, from the degree distributions for different increasing values of the threshold $y$, using the theory introduced by one of us[30]. In the multi-step setting, when the degradation occurs in a series of filtration-response steps, we observe that the disruption of $S_W$ is significantly postponed when compared to the case of no active response.

Figure 3 illustrates the percolation transition being significantly delayed as a result of the homeostatic response in a random graph and an empirical brain network[31,32]. The values obtained from direct numerical simulations of $S_W$ (colored markers) are well captured by the estimates (black× and +markers) obtained by the directed configuration model[30] supplied with the degree-distributions derived from the master equation (1).

Successive steps of the damage–response process may only occur in a system that quickly responds to an external perturbation. In some systems, synaptic scaling is reported to have a typical time of hours/days after external damage[20,33]. Therefore, we also analyze a single step of the process, mimicking a relatively slow response. As explained in the "Methods" section, in the single-step case, our model can be solved analytically provided the initial network does not have weight-degree correlations. Then even a single step introduces nontrivial correlations between weights and degrees (see "Methods" section for details), by which we lose the analytical tractability. Therefore letting $w_k(x, y = 0) = w(x, y = 0)$ be the initial weight distribution, the average weight after a single step of the damage–response process at threshold $y$, $\bar{w}(y) = \int_0^\infty x w(x, y)dx$, reads

$$\bar{w}(y) = \beta_1(y) + \beta_2(y)\left(\frac{F(y) - G(F(y))}{1 - F(y)}\right), \quad (3)$$

where $F(y) = \int_0^y w(x)dx$ is the initial cumulative weight distribution

$$\beta_1(y) = \frac{\int_y^\infty x w(x)dx}{1 - F(y)}, \quad (4)$$

$$\beta_2(y) = \frac{\int_0^y x w(x)dx}{F(y)}, \quad (5)$$

and $G(z) = \sum_{k \geq 0} l_k z^k$—the generating function of $l_k$.

The second term of Eq. (3) represents the contribution from the homeostatic response. As a matter of fact, one may verify that $\beta_1(y)$ coincides with the average weight in the case of no response (see the "Methods" section). On the other hand, the average degree after the damage is simply given by

$$\bar{k}(y) = \bar{k}(0)[1 - F(y)], \quad (6)$$

where $\bar{k}(0)$ stands for the initial average degree.

Figure 4 shows a very good agreement between our analytical predictions and stochastic simulations for the values of the average weight $\bar{w}(y)$ and the average degree $\bar{k}(y)$, in both synthetic and real networks.

Note that, by construction, our framework naturally takes into account the correlation between weights and degrees. In the "Methods" section we prove that, for a single instance of the damage–response process, the sign of this correlation is strongly affected by the underlying network topology, in particular, it is positive for scale-free networks, e.g., edges pointing to higher degree nodes have higher weights, while is negative for both random regular and Erdös–Rényi networks, e.g., edges pointing to higher degree nodes have lower weights.

From the results of Fig. 4, it is evident that $\bar{k}(y)$ is a decreasing function in $y$, while $\bar{w}$ is an increasing function in $y$. One can verify from equations (3) and (6) that this is true in general. In the Methods section, we show that the product of the two, that is $\bar{w}\bar{k}$, coincides with the network average strength $\bar{S}$. This quantity tends to be quasi-conserved for small values of $y$ in the absence of low-degree nodes (see Methods section), which is easily understood when we look back at the local rule (2): for each node $i$ the local strength is conserved only if at least one edge survives the filtration process, otherwise, the lost weight is not redistributed.

We can use the analytical estimate for $\bar{S}$ in order to approximate the leading eigenvalue of the network $\lambda_{max}$: in the case of weighted directed networks, $\lambda_{max}$ is approximated by $\overline{S_{in}S_{out}}/\bar{S}$[34,35], which approaches the average strength as we neglect in/out strength correlations.

Hence, by combining Eqs. (3) and (6) we can assess the behavior of $\lambda_{max}$ as well as other spectral properties. To this end we consider, as an example, the trace of the *communicability matrix*, i.e., the Estrada index (EE)[36], motivated by the recent applications in the fields of both brain networks[37,38] and traffic flows in cities[39]. Note that in this case, since both $\lambda_{max}$ and EE

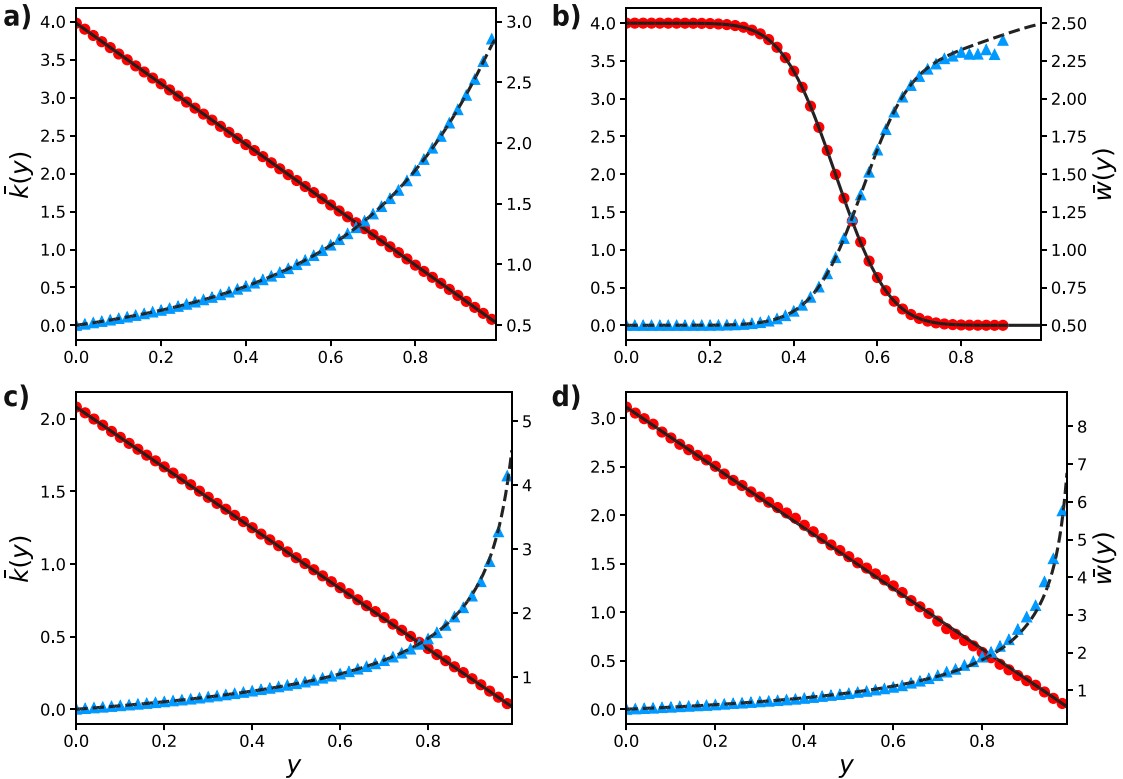

**Fig. 4 Average degree and average weight for single instances of the damage–response process.** Colored markers represent the outcome of a single stochastic simulation. Solid and dashed lines represent the analytical results for average weight and degree (derived in the Methods section). Red dots and continuous lines represent the average degree (left scale), while blue triangles and dashed lines represent the average weight (right scale). **a** Poisson network of $N = 25 \times 10^3$ nodes and uniform weights. **b** Random regular graph with $N = 25 \times 10^3$ nodes and normally distributed weights (same distribution of Fig. 2). **c** Scale-free network of $N = 25 \times 10^3$ nodes with power-law exponent $\gamma = 2.5$ and uniform weights. **d** Empirical network with uniform distributed weights (same network as in Fig. 3b).

depend on the weights, the effect of the damage/response process is observable also in the single-step case, while when we considered $S_W$, because the giant component is determined by the degree distribution only, the effect of the homeostatic response can only be seen in the multi-step case, for which we are sure that weight-degree correlations have formed.

Figure 5 shows that, similarly to the case of $S_W$, the homeostatic response has a clear effect of sustaining high values of both $\lambda_{max}$ and EE, compared to the case where no response is present. Moreover, in the case of a random regular network with uniformly distributed weights, the estimation of $\lambda_{max}$ by means of $\overline{S} = \overline{w}\overline{k}$ is quite accurate, both in the single-step case (panel a) and in the successive-steps case (panel b). Hence, we approximate the value of $EE = \sum_n e^{\lambda_n}$[36], by keeping the contribution of the largest eigenvalue plus the 0th order of the remaining part of the spectrum. Therefore, for this particular case we have $EE \sim N - 1 + e^{\lambda_{max}} \sim N - 1 + e^{\overline{w}\overline{k}}$. From Fig. 5, we see that this approximation is less accurate, but does not fail to capture the general behavior emerging from stochastic simulations.

## Discussion

To summarize, percolation-like processes that conceptualize random damage in networks have been since long viewed as prototypical models for complex systems resilience. However, systems that actively maintain their homeostatic response tend to have the means to respond and actively adapt to such damage. We have presented a theory that naturally extends the classical percolation framework to a more complex one that incorporates the principle of homeostatic plasticity. Being a natural extension of simple percolation, our framework is still a theoretical

abstraction but is not size-limited. Therefore, it is able to project the effects of local homeostatic mechanisms, similar to the ones observed in real biological systems at small sizes[16,40], at arbitrarily large scales, thus overcoming the practical limitations of laboratory experiments. By means of our model, it is possible to show that a simple local self-regulatory mechanism may significantly improve the large-scale functionality of the whole network compared to the case in which such a mechanism is absent. Our results reproduce the evolution of the joint weight-degree distribution of the network, which allows predicting the behavior of several global indicators of network structure and dynamics, such as the size of the largest connected component, the largest eigenvalue, and the Estrada index.

Overall our results provide a first mathematical framework for studying the link between local homeostatic plasticity rule in complex networks and its effect on the global functionality, and may also shed light on how the self-regulatory mechanisms observed in biological systems might be transferred to improve the resilience of human-designed infrastructures, for example, communication or transport networks, wherein it is reasonable to assume that homeostatic response might be adopted to mitigate external damage.

## Methods

**Notation and master equation**. We consider a general model for *homeostatic plasticity* in a random directed network wherein all edges have a weight $x > 0$. According to this process, the network updates the edges' weights in a response to filtration, which is a deterministic and simultaneous removal of all edges with a weight strictly below a certain threshold $y$. Such a response can be adopted, for example, to mitigate further damage due to filtration in the future[20,41]. We consider networks wherein each node has at most one in-edge and possibly many out-edges and adopt the following notation to analyze the evolution of the network structure:

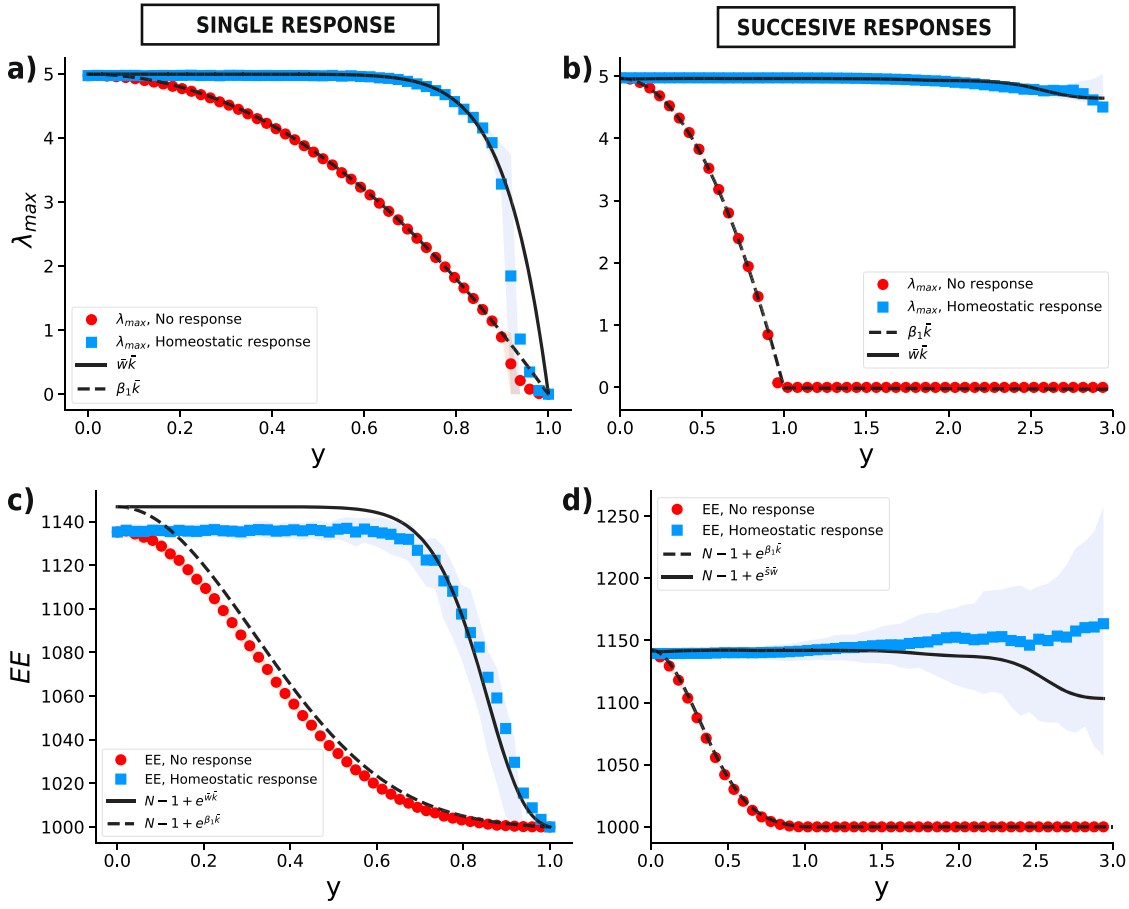

**Fig. 5 Leading eigenvalue and Estrada index for both single and multiple instances of the damage-response process.** Continuous and dashed black lines represent the values from our model in the case of homeostatic response and no response respectively, while colored markers and colored shaded areas represent mean values and interquartile ranges derived from stochastic simulations. We averaged the damage-response process over 200 realizations of a random regular network of $N = 10^3$ nodes and average degree $z = 10$ with uniform weights. **a**, **c** Single instances of the damage–response process. In this case $\bar{w}\bar{k}$ is simply given by combining Eqs. (3) and (6). **b**, **d** Successive instances of the damage–response process. Here, both $\bar{w}$ and $\bar{k}$ are computed at each step from the master equation (1). In every panel, $\beta_1$ is computed from Eq. (4).

- $p_k$ is the in-degree distribution. Related to this quantity is $\bar{k} := \sum_{k \geq 0} k p_k$—the average degree, and $l_k := k p_k / \bar{k}$—the *excess degree* distribution, which denotes the probability of uniformly at random picking an edge sitting on a node of in-degree $k$.
- $w_k(x)$ is the probability density function such that $\mathbb{P}(w > x | deg = k) = \int_x^\infty w_k(\tau) d\tau$ is the probability of picking an in-edge with a weight greater than $x$, *given that* it points to a node with in-degree $k$. Note that plasticity may result in weight-degree correlation (see the last section), therefore $\mathbb{P}(w > x | deg = k) \neq \mathbb{P}(w > x)$.
- $f_k(x) := l_k w_k(x)$ is the joint probability density function, corresponding to the event of picking an edge with weight $w \in [x, x + dx]$ pointing to a node of in-degree $k$. Let us denote the marginal distributions of $f_k(x)$ as

$$l_k := \int_0^\infty f_k(x) dx,$$

$$w(x) := \sum_{k \geq 0} f_k(x).$$

Since the response mechanism involves the weights of in-edges only, we refer to the in-degree as $k$. Response of degree-weight density function $f_k^0(x)$ to filtration with threshold $y > 0$ is given by

$$f_k^1(x) = \mathcal{A} f_k^0,$$

where

$$\mathcal{A} f_k(x) = \sum_{n \geq k} \mathcal{D}_{k,y}(f_n(x)) * \mathcal{R}_{k,y}(f_n(x)), \quad (7)$$

In Eq. (7), the contribution to the $k$th-degree nodes after filtration comes from nodes with degree $n \geq k$. For these nodes, *Damage operator* $\mathcal{D}_{k,y}$ conceptualizes the

effect of filtration on the node degrees and edge weights, whereas the *Response operator* $\mathcal{R}_{k,y}$ represents the response to filtration by increasing weights of surviving edges. The convolution operation $*$ is defined by $(f * g)(x) := \int_{-\infty}^\infty f(\tau) g(x - \tau) d\tau$.

**The Damage operator.** The *Damage* on the network is represented by a filtration process in which all the edges with weight below a fixed threshold $y$ are removed. In order to get the explicit form of $\mathcal{D}_{k,y} f_n(x)$ we compute the damaged excess degree distribution $l_k(y)$ and the damaged weight distributions $w_k^D(x, y)$. The latter is simply given by the original weight distribution $w_k(x, 0)$ cut and renormalized

$$w_k^D(x, y) = \frac{w_k(x, 0) \theta(x - y)}{1 - F_k(y)}, \quad (8)$$

where $F_k(y)$ indicates the cumulative distribution function of $w_k(x, 0)$, defined as

$$F_k(y) = \int_0^y w_k(x, 0) dx. \quad (9)$$

On the other hand by definition $l_k(y) = k p_k(y) / \bar{k}(y)$, hence we just need to compute the damaged degree distribution. For every node of degree $n$, each of the $n$ in-edges may be removed with probability $F_n(y)$, therefore we have

$$p_k(y) = \sum_{n \geq k} B(k, n, 1 - F_n(y)) p_n(0), \quad (10)$$

where $B(k, n, 1 - F_n(y))$ denotes a binomial distribution with parameters $(k, n, 1 - F_n(y))$.

The average degree after damage is then given by

$$\bar{k}(y) = \bar{k}(0)(1 - \langle F_n(y) \rangle_0), \quad (11)$$

were we defined $\langle f_k \rangle_y := \sum_{k \geq 0} f_k l_k(y)$.

With this notation one can rewrite Eq. (10) in terms of the excess degree distribution $l_k$, which, combined with Eq. (8), finally yields the explicit form of the Damage operator

$$\mathcal{D}_{k,y} f_n(x) = \frac{\frac{k}{n} B(k, n, 1 - F_n(y)) \theta(x - y)}{(1 - \langle F_n(y) \rangle_0)(1 - F_n(y))} f_n(x, 0). \tag{12}$$

**The Response operator**. Here, we present the explicit form for the response operator $\mathcal{R}_{k,y}$ introduced in Eq. (7) for our particular case in which the response is locally specified by the rule (2). In terms of a probability distribution, similarly to Eq. (8) we first compute the distributions of the deleted weights $w_k^c(x, y)$ which is given by

$$w_k^c(x, y) = \frac{w_k(x, 0) \theta(y - x)}{F_n(y)}. \tag{13}$$

From Eq. (13) we get the explicit form of $\mathcal{R}_k$ by considering the distribution associated with the sum of $n - k$ deleted edges scaled by a factor $k$

$$\mathcal{R}_{k,y} f_n(x) = [k w_n^c(kx, y)]^{*n-k} = \left[ \frac{w_n(kx, 0) \theta(y - kx)}{F_n(y)/k} \right]^{*n-k}. \tag{14}$$

Overall we can now write the explicit form of the master equation (7) for this process

$$f_k(x, y) = \sum_{n \geq k} \left[ \frac{\frac{k}{n} B(k, n, 1 - F_n(y)) \theta(x - y)}{(1 - \langle F_n(y) \rangle_0)(1 - F_n(y))} f_n(x) \right] * \left[ \frac{w_n(kx, 0) \theta(y - kx)}{F_n(y)/k} \right]^{*n-k}. \tag{15}$$

By repetitively applying Eq. (15) at increasing values for $y$ we get the evolution of $f_k(x, y)$ for a multi-step process. The plots presented in Figs. 2 and 3 of the Results section are derived by computing the marginals of $f_k(x, y)$ at each step. In particular, the in-degree distribution is derived from $l_k = \int_0^\infty f_k(x) dx$, combined with Eq. (11) and the definition $p_k = \bar{k} l_k / k$. On the other hand, the weight distribution is given by $w(x, y) = \sum_k f_k(x, y)$. Finally, the out-degree distribution is simply given by directly applying Eq. (10) to the initial $p_k^{out}$ at any value for $y$, since the response mechanism involves weights of in-edges only, hence no additional correlation between out-degrees and edge weights is introduced in the process. With both in and out-degree distributions, we then estimate the size of the weakly largest connected component, $S_W$, for a directed configuration model[30], as threshold $y$ is increased.

**Single-step from an initial state with no weight-degree correlation**. Here, we show that if the initial configuration is given by a network with no weight-degree correlation, the outcome of a single damage-response step is analytically tractable.

Let $w_k(x, 0) = w(x, 0)$, so that the initial weight-degree distribution is given by $f_k(x, 0) = l_k(0) w(x, 0)$ and the only cumulative weight distribution is given by $F(y) = \int_0^y w(x) dx$.

Under these assumptions, Eq. (15) can be rewritten as

$$f_k(x, y) = \sum_{n \geq k} \frac{k}{n} \binom{n}{k} (1 - F(y))^{k-1} F(y)^{n-k} l_n(0) \left[ \frac{w(x, 0) \theta(x - y)}{1 - F(y)} \right]$$
$$* \left[ \frac{w(kx, 0) \theta(y - kx)}{F(y)/k} \right]^{*n-k}, \tag{16}$$

where we highlighted in the square brackets the damaged weight distribution and the response operator. Let the convolution of the two be $W_{k,y}(x)$. Its Laplace

transform, defined as $\mathcal{L}(g(x)) = \int_0^\infty g(x) e^{-sx} dx$, reads

$$\mathcal{L}(W_{k,y}(x)) = \left[ \frac{\int_y^\infty w(x, 0) e^{-sx} dx}{1 - F(y)} \right] \left[ \frac{\int_0^y w(x, 0) e^{-\frac{sx}{k}} dx}{F(y)} \right]^{n-k}.$$

From the properties of the Laplace transform, it is then straightforward to obtain

$$\int_0^\infty x W_{k,y}(x) dx = \beta_1(y) + \left( \frac{n - k}{k} \right) \beta_2(y), \tag{17}$$

where we defined

$$\beta_1(y) = \frac{\int_y^\infty x w(x, 0) dx}{1 - F(y)} \qquad \beta_2(y) = \frac{\int_0^y x w(x, 0) dx}{F(y)}. \tag{18}$$

After having introduced the two functions above, we can conveniently write $\bar{w}(y)$ as

$$\bar{w}(y) = \beta_1(y) + \beta_2(y) \left( \frac{F(y) - G(F(y))}{1 - F(y)} \right), \tag{19}$$

where $G(z) = \langle z^k \rangle_0 = \sum_k l_k(0) z^k$ is the generating function of $l_k(0)$.

The results were obtained by testing the predictions of Eqs. (19) and (11) on several networks (both synthetic and real) are shown in Fig. 4 of the main article.

**Average strength quasi-conservation**. From Eq. (2) it is easily verified that the *local* strength is conserved if at least one edge survives. Here we show that from Eq. (19), one can verify that under certain assumptions the average strength of the whole network $\bar{S}$ is quasi-conserved. We start by showing that $\bar{S} = \bar{w} \bar{k}$. By definition, the in/out strength of a node $S_i$ is defined by the sum of all the in/out weights. The strength distribution $P(S = x)$ (either in or out) can therefore be written as

$$P(S = x) = \sum_k p_k [w_k(x)]^{*k}.$$

Note that we allow the presence of weight-degree correlation for the most general case. However, we do not allow the correlation between weights belonging to the same neighborhood.

The average strength, $\bar{S}$, therefore reads

$$\bar{S} = \sum_k p_k \mathbb{E}[w_k^{*k}] = \sum_k k p_k \bar{w}_k = \bar{k} \sum_k l_k \bar{w}_k = \bar{k} \bar{w}.$$

Let us now consider Eq. (19). We note that since $F(y) \in [0, 1]$, the contribution given by $G(F(y))$ can be neglected if low degree nodes are not present and $F(y)$ is small enough. Under these assumptions, $F^k(y) \ll 1$, i.e., the probability that for a node of degree $k$ there are no surviving edges after damage, is very small. Therefore Eq. (19) can be approximated by

$$\bar{w}(y) \simeq \frac{\bar{w}(0)}{1 - F(y)} = \frac{\bar{w}(0)}{\bar{k}(y)/\bar{k}(0)},$$

where we used Eq. (11) with $F_n(y) = F(y)$.

The above equation then implies

$$\bar{w}(y) \bar{k}(y) \simeq \bar{w}(0) \bar{k}(0), \tag{20}$$

meaning that $\bar{S}$ is quasi-conserved by the response mechanism.

**The emergence of weight-degree correlation**. In this last section, we show that a single instance of the damage response process generates either positive or negative

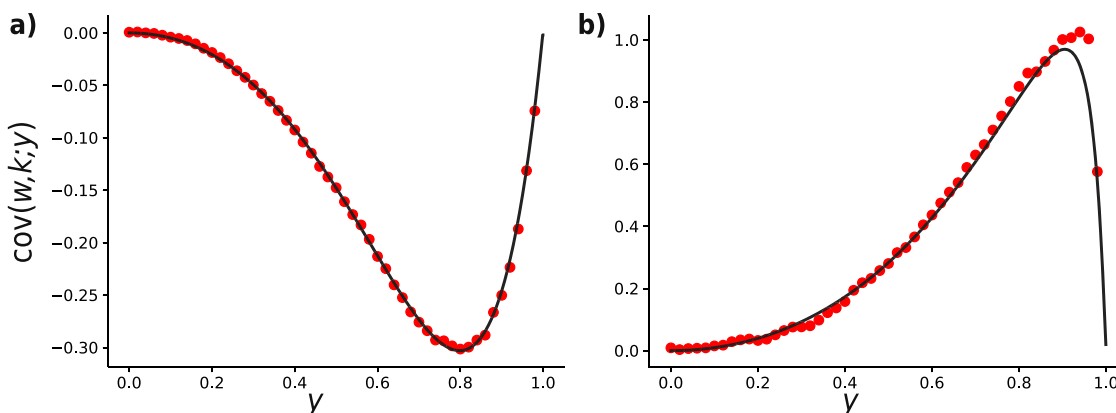

**Fig. 6 Weight-degree covariance as a function of threshold y for a single instance of the damage–response process.** The initial configurations are given by: **a** a $N = 25 \times 10^3$ nodes random regular graph with uniform weight distribution and **b** a $N = 25 \times 10^3$ nodes scale-free network with exponent $\gamma = 2.5$, with uniform weight distribution (right panel). In both plots red markers represent the results from a single stochastic simulation, while continuous black lines show the analytical values from the theory. As we can see in the regular graph case the covariance is always negative, while the opposite happens in the scale-free network case.

weight-degree correlations. Since the sign of the correlation coefficient is determined by the one of covariance, we can focus ourselves on the covariance only. By definition $\mathrm{cov}(k,w) = \mathbb{E}[kw] - \mathbb{E}[k]\mathbb{E}[w]$, which in our notation reads $\langle k\bar{w}_k \rangle_y - \langle k \rangle_y \bar{w}(y)$. Note that in our notation $\langle k \rangle$ is not the average degree, but it's the first moment of the excess-degree distribution. By using again Eq. (17), one can obtain

$$\langle k\bar{w}_k \rangle_y = \beta_1(y)\langle k \rangle_y + \beta_2(y)(\langle k \rangle_0 - \langle k \rangle_y), \tag{21}$$

which combined with the previous results yields

$$\mathrm{cov}(w,k;y) = \beta_2(y)\left( \langle k \rangle_0 G(F(y)) - \frac{F(y)(1 - G(F(y)))}{1 - F(y)} \right). \tag{22}$$

Note that $\beta_2(y) = 0 \Rightarrow \mathrm{cov}(k,w;y) = 0$, meaning that in the case of no response no additional weight-degree correlation is introduced, as expected.

From the last equation, we derive the condition

$$\mathrm{cov}(w,k;y) \geq 0 \iff G(F(y)) \geq \frac{F(y)}{\langle k \rangle_y}. \tag{23}$$

One can verify the condition above is never satisfied for either Erdös–Rényi or Random Regular networks (for the case of Random Regular networks although the strict inequality is never satisfied, the equality holds for extreme case of constant degree $z = 1$), while for Scale-Free networks the opposite result holds.

In Fig. 6, we show the comparison between the analytic theory and stochastic simulations on a random regular network and a scale-free one, both with uniform weight distribution.

## Data availability
The authors declare that all data supporting the findings of this study are available within the article.

## Code availability
The code for reproducing the main findings of the article is available at the GitHub repository: https://github.com/grapisar/PercHomeostatic.

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

## Acknowledgements
G.R. acknowledges the financial support of the Barcelona Supercomputing Center, A.A. acknowledges support by Ministerio de Economía y Competitividad (Grants No. PGC2018-094754-B-C21), Generalitat de Catalunya (Grant No. 2017SGR-896), Universitat Rovira i Virgili (Grant no. 2019PFR-URV-B2-41), ICREA Academia, and the James S. McDonnell Foundation (Grant no. 220020325). I.K. acknowledges the kind hospitality of the Alephsys lab at Universitat Rovira i Virgili.

## Author contributions
G.R. and I.K. developed the analytical theory and performed numerical simulations, G.R., I.K., and A.A. designed the project, interpreted results, and wrote the paper.

## Competing interests
The authors declare no competing interests.
