## [Peer Review File · Nature Communications]

REVIEWER COMMENTS

Reviewer #1 (Remarks to the Author):

The world is not perfect, neither are networks. And even if they are, many processes can damage them. The authors transfer the concept of homeostasis of neural networks that results in a healthy "homeostatic" degree of activity and functional connectivity under stress to percolation and its relation to resilience. Network damage is represented by a filtration process in which all the edges with weight below a fixed threshold are removed. Similar to an evolutionary feature in the real world, the authors then apply a Response Operator that redistributes the weights, to maintain connectivity, mimicking homeostasis.

The mathematics is nontrivial because they study directed networks and two nontrivial operators on them. Yet, all analytical approximations more or less perfectly match their network instances. The analysis is well carried out and impressive. For example, they compute the single damage-response step analytically, if the initial configuration is given by a network with no weight-degree correlation. In addition, they study Erdos-Renyi, Random Regular networks and scale-Free networks analytically, and leading eigenvalue and Ernesto Estrada Index to quantify the damage-response process and resilience, suitably.

I recommend publication, with possible minor changes as suggested below.

The authors write

"...that resembles paradigmatic critical transitions in epidemic spreading, biological networks, traffic and transportation systems."

I would (better) clarify already in the abstract but latest in the introduction when "resemblance" of critical transitions is exact for SI/SIS/SIR and co and when not. I recommend to better motivate when percolation actually describes transitions in less "paradigmatic" but more real systems in our real world. It is also not wrong to cite some papers on mappings of percolation to contact processes or epidemic (local) spreading, e.g. Grassberger Phys A of 1985, or be a bit more precise. The readership may love it.

There is more literature on rewiring/re-weighting networks and their consequences, e.g., published in Nature Communications, e.g.,

<https://www.nature.com/articles/ncomms9412>

<https://www.nature.com/articles/ncomms10441>

<https://www.nature.com/articles/s41467-021-21824-x>

<https://www.nature.com/articles/ncomms1163>

Those literature pieces (or part of it) may help to provide other examples where homeostasis, or processes similar to that concept, in particular redistribution processes, have been studied in the context of resilience, in particular in ecosystems and aviation. If the authors find that all those papers plus references therein have nothing to do with the author's novel idea, what might be mistaken, maybe they want to give it a try and explain the differences in the introduction. Again: The readership may love it.

Of course, there are at least two paper that must be cited, in the general context, on percolation in (living) neural systems:

Percolation in Living Neural Networks,

Ilan Breskin, Jordi Soriano, Elisha Moses, and Tsvi Tlusty

Phys. Rev. Lett. 97, 188102, 2006

and the(ir) "follow up" of 2008 that appeared in PNAS:

<https://www.pnas.org/content/105/37/13758>

If the authors like it, here is what 2020 offers,
<https://arxiv.org/abs/2001.08669> (why not checking out the references therein?).

The sentence

"resilience is defined BY the interplay between the network structure and link weights"
is perhaps a bit too colloquial.

I recommend not only to refer to ref 22 but to explain what
the largest *weakly* connected component is.

There are more peaks in Fig2, for homeostasis. Ok. But why is that good or important?
One could *better/more* explain that in the main text, in the context of resilience and its
mechanisms.

Reviewer #2 (Remarks to the Author):

I studied the manuscript "A theory for percolation in active networks that maintain homeostasis" by
Rapisardi, Kryven, and Arenas.

The authors study the problem of quantifying the robustness and retuning the resilience of active
networks (taken as prototypical models of biological networks of neural cells) after a damage as
occurred at the level of the (pre)synaptic connections of the connectome.

The first problem (the robustness) is tackled via the theory of link-percolation
induced by a multistep filtration process and it is a known result.

The second issue (retuning the network links to cope with the damage) is studied and resolved
through a mechanism of homeostatic regulation, which shows a prominent role in maintaining the
network highly resilient even for severe damage levels (high filtration threshold). This is a sound new
result.

Overall, I enjoyed reading this manuscript: the problem is cool, the theory is well developed, and the
narrative is catchy. Therefore I am convinced that it deserves publication in Nature Communications
as is.

I am curious to know if there is an answer to the following few questions of mine, that came up while
reading the paper:

1) What happens when inhibitory synapses are present?, i.e., when negative weights are added to the
model. Every brain has both excitatory (already present in the paper) and inhibitory synapses (absent
in your model) and thus a model aiming at explaining brain function (and malfunction) cannot be
satisfactory without incorporating both positive and negative synaptic weights.

2) How do your results change in the case of mode-like percolation instead of the link-percolation case
you address?

3) Wouldn't it be more clever for the brain to readjust only the subcritical weights, i.e. the weights
slightly under threshold, instead of retuning the whole input set of damaged nodes?

4) What about the resilience of the strongly connected component G ? That is, how does the percolation plot of $G(y)$ look like for no-response VS homeostatic response? (Basically the plot in fig 3 for $S \rightarrow G$).

Reviewer #3 (Remarks to the Author):

A question of interest in various fields is how the underlying networks adapt to changes. In this paper, a model is considered in which the loss of network connections is compensated by increasing the weights of the remaining connections. This is implemented using a simple local rule, which allows the calculation of percolation properties and leading eigenvalues. As expected, the rescaling of connection weights reduces the damage caused by the lost connections. The analysis appears sound and the presentation is transparent. The perspective on potential applications presented in the paper (e.g. to smart infrastructures) is also interesting.

However, I feel that the model and results are a bit too simple and unsurprising to allow general conclusions or concrete applications. It is indicated in the paper that the model is inspired by synaptic scaling in neurons, but the results are not validated against data in a specific context. The significance of the work is then exclusively based on the model and the results obtained from the model.

Minor comments:

Key concepts (such as homeostatic strategies and active networks) should be defined the first time they are used.

Most figures being clearly described, but some captions could be more specific about the network models and weight distributions used (including their mean and variance when applicable).

The following statement is unclear and its relation to fig 4 should be explained: "the sign of this correlation is strongly affected by the underlying network topology, in particular, it increases for scale-free networks, while it decreases for both random regular and Erdos-Renyi networks"

Response to Referee 1

Comment/question: The authors write "...that resembles paradigmatic critical transitions in epidemic spreading, biological networks, traffic and transportation systems." I would (better) clarify already in the abstract but latest in the introduction when "resemblance" of critical transitions is exact for SI/SIS/SIR and co and when not. I recommend to better motivate when percolation actually describes transitions in less "paradigmatic" but more real systems in our real world. It is also not wrong to cite some papers on mappings of percolation to contact processes or epidemic (local) spreading, e.g. Grassberger Phys A of 1985, or be a bit more precise. The readership may love it.

Reply: Thanks for the nice suggestion. We have done our best to better motivate this point in the revision of the manuscript. In the abstract now it reads: "Some biological systems, such as networks of neural cells, actively respond to percolation-like damage, which enables these structures to maintain their function after degradation and aging. Here we present a theory for studying percolation in networks that actively respond to link damage by adopting a mechanism resembling synaptic scaling in neurons. The theory explains critical transitions in such active networks and shows that these structures are more resilient to damage as they are able to maintain a stronger connectedness and ability to spread information." Accordingly, in the last part of the introduction, we add: "The most notable example refers to explaining the spreading of a disease driven by the Susceptible-Infected-Recovered contact process. Percolation in a random network is identical to this process when the infectious time is constant[9] and leads to a useful approximation otherwise[10]. Percolation has been also used to understand the early stages of formation of the brain by probing how resilient are interconnected neuronal cultures in vitro[11,12]"

*Comment/question: There is more literature on rewiring/re-weighting networks and their consequences, e.g., published in Nature Communications, e.g.,
<https://www.nature.com/articles/ncomms9412>
<https://www.nature.com/articles/ncomms10441>
<https://www.nature.com/articles/s41467-021-21824-x>
<https://www.nature.com/articles/ncomms1163>*

Those literature pieces (or part of it) may help to provide other examples where homeostasis, or processes similar to that concept, in particular redistribution processes, have been studied in the context of resilience, in particular in ecosystems and aviation. If the authors find that all those papers plus references therein have nothing to do with the author's novel idea, what might be mistaken, maybe they want to give it a try and explain the differences in the introduction. Again: The readership may love it.

Indeed, pertinent references that we probably overlooked. All references have been added, thanks for your input.

Comment/question: Of course, there are at least two paper that must be cited, in the general context, on percolation in (living) neural systems: Percolation in Living Neural Networks, Ilan Breskin, Jordi Soriano, Elisha Moses, and Tsvi Tlusty Phys. Rev. Lett. 97, 188102, 2006 and the(ir) "follow up" of 2008 that appeared in PNAS:

<https://www.pnas.org/content/105/37/13758> If the authors like it, here is what 2020 offers, <https://arxiv.org/abs/2001.08669> (why not checking out the references therein?).

Indeed. Thanks for the suggestion, the references to the first two articles are now included.

Comment/question: The sentence "resilience is defined BY the interplay between the network structure and link weights" is perhaps a bit too colloquial.

Thanks for your comment, we have made the sentence less colloquial, this sentence now reads: "both the structure and link weights contribute to network resilience".

*Comment/question: I recommend not only to refer to ref 22 but to explain what the largest *weakly* connected component is.*

We agree on this point, the largest weakly connected component is now explicitly defined the first time it appears on the manuscript, in page 7.

*Comment/question: There are more peaks in Fig2, for homeostasis. Ok. But why is that good or important? One could *better/more* explain that in the main text, in the context of resilience and its mechanisms.*

Thanks for raising this point. In Figure 2 we demonstrate that multimodality promotes resilience. While the total weight is conserved, adding an additional mode at larger weights is a more economical way of weight redistribution than making the initial unimodal distribution 'broader'.

The figure also shows that, even though filtration with adaptive response may cause emergence of multimodality in the weight distribution, our analytical theory is capable to explain such complex patterns, as we already explicitly pointed out in the manuscript. This also suggests that a simpler equation that tracks only several first moments instead of the whole weight distribution, will perhaps not be suitable to study this phenomenon.

Response to Referee 2

Comment/question: 1) What happens when inhibitory synapses are present?, i.e., when negative weights are added to the model. Every brain has both excitatory (already present in the paper) and inhibitory synapses (absent in your model) and thus a model aiming at explaining brain function (and malfunction) cannot be satisfactory without incorporating both positive and negative synaptic weights.

This is a very interesting point, thanks for raising it up. Before addressing it, we would like to state that our model is not aimed to reproduce specifically the brain functionality, but it is rather inspired by a particular phenomenon, the homeostatic plasticity, that indeed takes place in our brains. Due to its generality, the model could be potentially applied to different real-life networks that self-regulate in a similar way, but again, a detailed description of the brain's functioning is far beyond of our scientific capabilities, and definitely not the main scope work.

Nevertheless, as it is a basic model, it can be easily expanded to account for additional features. Even though the literature on synaptic scaling is more focused on the role of excitatory synapses, there is evidence for similar mechanisms to hold on inhibitory synapses

as well

- <https://pubmed.ncbi.nlm.nih.gov/11850460/>
- <https://pubmed.ncbi.nlm.nih.gov/21123568/>
- <https://www.ncbi.nlm.nih.gov/pmc/articles/PMC3396123/>

A possibility could be, therefore, that of considering a multiplex network structure, the first layer being defined by excitatory synapses (positive weights) and the second by inhibitory ones (negative weights), in which the same local conservation rule holds separately in the two layers. Applying our model independently to each layer is feasible and the main insights are essentially the same. However, the entanglement of excitatory and inhibitory signals in the same framework resisted our mathematical abilities so far. We are deceived of not providing a better reply to this important question. Nevertheless, we take good note on this point to enlarge, in the next future, the possibilities of our theory beyond the current stage.

Comment/question: 2) How do your result change in the case of node-like percolation instead of the link-percolation case you address?

Thank you for the question. To consider a node-percolation setting we would have to reformulate the whole model. Since the homeostatic mechanism (or synaptic scaling) involves the links weights only, the aggregation of these weights into the strength of the node will provoke a totally different outcome, worth to explore in a dedicated manuscript.

Comment/question: 3) Wouldn't be more clever for the brain to readjust only the subcritical weights, i.e. the weights slightly under threshold, instead of retuning the whole input set of damaged nodes?

Thank you for your comment. Mathematically speaking, this would disturb the distribution of the weights at each node, and hence would likely affect the functioning of the (neural) network, e.g. the by affecting the leading eigenvalue. Even from the purely structural point of view, updating only the weakest links may help in the short run but will also narrow the weight distribution and hence will likely lead to a catastrophic collapse later on. In other words: stronger links are also very important ones, and need to be taken care of too.

Comment/question: 4) What about the resilience of the strongly connected component G ? That is, how does the percolation plot of $G(y)$ look like for no-response VS homeostatic response? (Basically the plot in fig 3 for $S \rightarrow G$).

From a “microscopic” description level, the process we model has the only effect of improving links resilience to increasing damage, hence for the *strong* giant component G we expect the same phenomenology observed for the *weak* component S , that is a noticeable delay in the percolation transition. Then, since G is a set nodes defined by more stringent rules, it is to expect also that degradation occurs earlier compared to S , in both the cases, i.e. no response and homeostatic response.

To validate this claims, we performed new computer simulations of the model, and here we append a simple plot from a numerical simulation on a random regular network of in-/out-degree $z = 4$ and size $N = 10^4$ nodes, with uniformly distributed weights.

Figure 1. Evolution of both weak giant component (left panel) and strong giant component (right panel) on a random regular network with in-/out-degree $z = 4$, $N = 10^4$ and uniform distributed weights.

Response to Referee 3

Comment/question: Key concepts (such as homeostatic strategies and active networks) should be defined the first time they are used.

Thanks for the useful suggestion. We rewrote the abstract in simpler words, so that the meaning of active network becomes clearer. We also defined the homeostatic response in the introduction.

Comment/question: Most figures being clearly described, but some captions could be more specific about the network models and weight distributions used (including their mean and variance when applicable).

Thanks for the suggestion. We have added details in the captions about the used weight distributions and algorithm used to generate the synthetic networks.

Comment/question: The following statement is unclear and its relation to fig 4 should be explained: “the sign of this correlation is strongly affected by the underlying network topology, in particular, it increases for scale-free networks, while it decreases for both random regular and Erdos-Renyi networks”

Thanks for the comment, we agree that the phrasing is not clear. The sentence now reads: ”In the Methods section we prove that, for a single instance of the damage-response process, the sign of this correlation is strongly affected by the underlying network topology, in particular it is positive for scale-free networks, *e.g.* edges pointing to higher degree nodes have higher weights, while is negative for both random regular and Erdős-Rényi networks, *e.g.* edges pointing to higher degree nodes have lower weights.”

REVIEWER COMMENTS

Reviewer #2 (Remarks to the Author):

I read the reply of the Authors to my comments. The rebuttal to the concerns I've raised acknowledges that those concerns of mine are indeed important issues to be addressed, although only in a followed up paper. I can take this answer as satisfactory. This I remain convinced that the paper deserves publication.
Best regards.

Reviewer #3 (Remarks to the Author):

The authors have revised some parts of the presentation, but I do not think that my main concerns have been addressed. They did address the minor points I raised but did not address (or even quote in their responses) the major ones. I will reiterate more explicitly:

1. The problem considered is somewhat trivial. The paper only studies what happens when a threshold is applied to the links of a network and the weights of the remaining links are increased. Very simple problems can lead to important conclusions, but in this case, I think the conclusions are of limited value.
2. The problem is not motivated. The paper mentions many processes and systems, like epidemics, aging, neural cells, ecological networks, transportation, etc. However, none of the cases mentioned is close to being described by the type of process considered in the paper, and no concrete connection is established with any real-world case.
3. "Homeostasis" is a misdenomination of what is described in the paper. The process in the paper is definitely not close to what this concept describes in biological systems. In mathematics, there is ample literature on the mathematical formulation of homeostasis in various contexts, including networks. But the process in this paper shows no relation with that literature either (and that literature is not cited in the paper).
4. Lack of a significant implication. Because the problem considered is too simple to be generalized and too abstract to relate to real-world systems, the implications that follow from it are also very limited.

I do think the initial idea in this paper has good potential and deserves to be studied. I also think the numerical results and mathematical calculations are correct. My objection is that the actual work and results presented in the paper do not go nearly far enough to justify publication at this state.

My opinion would be different if the paper validated the model against at least one real-world network, or if it established sufficiently general results on network adaptation to damage (not limited to thresholding) that could constitute a new addition to network theory.

Here are the point-to-point replies for each report:

The authors have revised some parts of the presentation, but I do not think that my main concerns have been addressed. They did address the minor points I raised but did not address (or even quote in their responses) the major ones. I will reiterate more explicitly:

We apologise to the referee, it has been our intention to address all points in the former review. We think that part of the apparent misunderstanding is due to some terminology in our manuscript, that may be interpreted in a different way than what we originally intended. For instance, we have realised that our definition of homeostasis is different from what Referee seem to have in mind. Now, after this ambiguity have been clarified (see the answer to point 3), we are convinced that the manuscript has become much clearer. We replaced the word *homeostasis* with the more specific phrasing *homeostatic plasticity* and added more discussion in the Introduction section to clarify this potential ambiguity. The title of the manuscript has also been changed to *A theory for percolation in networks with local homeostatic plasticity*.

We hope that after these changes the Referee will be satisfied with the new version.

1. The problem considered is somewhat trivial. The paper only studies what happens when a threshold is applied to the links of a network and the weights of the remaining links are increased. Very simple problems can lead to important conclusions, but in this case, I think the conclusions are of limited value.

We do agree with the referee statement in a general context, but we disagree with respect to the value of our conclusions. Processes that dynamically change the structure of a network in time is a new and expanding area of research. So far, on the theoretical side, most attention has been focused on percolation-like processes that have found many of applications. Here we shift the paradigm and consider networks that actively restructure after links are removed. This process is governed by a more complex, non-linear operator. Even though the formulation of the problem is very simple, on the computational side (both numerical and analytical) is quite challenging, since it involves infinite series with arbitrarily large convolutions. As we show in the paper, while the master equation we present is very general and straightforward, the window for an analytical solution is very narrow.

As for thresholding, it is a simple but yet important technique for studying the resilience properties of weighted networks that naturally extends bond percolation on unweighted ones. In fact, bond percolation is a special case of applying an increasing threshold to a weighted network where link weights are uniformly distributed. Additionally, in our model thresholding couples removing links with link weights, which adds an additional level of complexity. The other reason why we consider thresholding comes from applications in neuroscience (Martijn P. van den Heuvel, "Proportional thresholding in resting-state fMRI functional connectivity networks and consequences for patient-control connectome studies: Issues and recommendations", <https://www.sciencedirect.com/science/article/pii/S105381191730109X>).

The conclusions of our manuscript are crucial to understand theoretically the evolution of the strength distribution in weighted networks that endow the above mentioned plasticity mechanism, and sheds light to many interdisciplinary problems where such an analytical evolving distribution of weights is crucial to understand the evolution of the particular network under analysis, and changes in its properties.

Very simple problems can lead to important conclusions, but in this case, I think the conclusions are of limited value.

We think our results are relevant because they allow to scale the effect of a simple local rule (inspired by the observed behavior of neurons) to arbitrarily large systems, hence going beyond the physical limitations of both synthetic and real experiments. Moreover, even if our theory does not describe in detail a particular system, it surely represents a first model that extends percolation-like processes to a more complex framework which takes into account both damage and local self-regulatory responses.

2. The problem is not motivated. The paper mentions many processes and systems, like epidemics, aging, neural cells, ecological networks, transportation, etc. However, none of the cases mentioned is close to being described by the type of process considered in the paper, and no concrete connection is established with any real-world case.

Our model is a theoretical abstraction, which in any moment wants to be a faithful representation of neither brain networks, nor ecological networks, nor traffic jams, just like simple percolation is not meant to be a framework oriented to study any scenario in particular, but it rather represents a standard benchmark for understanding general resilience properties of a network, which many times is the scope of theoretical analysis.

That being said, the applicability of standard percolation is naturally restricted to networks that takes damage without reacting. Trying to overcome this fact is a known difficult challenge, since in principle there are countless ways of implementing a network response. As a reasonable candidate to represent at least a family of responsive systems, we took inspiration from synaptic scaling, which is a type a homeostatic response, well documented in real experiments for decades now. For example:

- G.G. Turrigiano & S.B. Nelson "Homeostatic plasticity in the developing nervous system" 2004, <https://www.nature.com/articles/nrn1327>.
- S.Teller et al. "Spontaneous Functional Recovery after Focal Damage in Neuronal Cultures" 2019, <https://www.eneuro.org/content/7/1/ENEURO.0254-19.2019>.
- E. Estévez-Priego, et al. "Functional strengthening through synaptic scaling upon connectivity disruption in neuronal cultures" 2020, <https://www.ncbi.nlm.nih.gov/pmc/articles/PMC7781611/>.
- G. S. Helena, "Recovery of neuronal networks after physical damage" 2021, <http://diposit.ub.edu/dspace/handle/2445/180289>.

3. "Homeostasis" is a misdenomination of what is described in the paper. The process in the paper is definitely not close to what this concept describes in biological systems. In mathematics, there is ample literature on the mathematical formulation of homeostasis in various contexts, including networks. But the process in this paper shows no relation with that literature either (and that literature is not cited in the paper).

Thank you for pointing out this. Indeed, there are several possible interpretations. According with Cambridge Dictionary, the definition of homeostasis (a term borrowed from biology) is as follows: 'the ability or tendency of a living organism, cell, or group to keep the conditions inside it the same despite any changes in the conditions around it, or this state of internal balance'. We are using the term in a close relation to its original definition, and we are not using the term homeostasis as in the dynamical systems literature. Several fundamental papers in the dynamical systems literature, as for example:

1. Martin Golubitsky and Ian Stewart, "Homeostasis, singularities, and networks" Journal of Mathematical Biology
<https://link.springer.com/article/10.1007/s00285-016-1024-2>
2. Martin Golubitsky and Ian Stewart, "Homeostasis with Multiple Inputs" SIAM JADS
<https://epubs.siam.org/doi/10.1137/17M115147X>
3. Golubitsky, Martin (et al.), "Input-Output Networks, Singularity Theory, and Homeostasis"
<https://www.springer.com/gp/book/9783030512637>,

use homeostasis addressing a partially a different meaning. In the aforementioned literature,

homeostasis relates to a stable fixed point of a system of ODEs. Even though this formalism is in principle very general and had lead to remarkable findings through the years, we think it does not apply to our case for several reasons, the first being that we do not use differential equations, nor we introduce any concept of equilibrium. We describe emerging homeostatic properties by projecting the effect of a local conservation rule (observed in biological systems) to a global scale, thanks to the use of a probabilistic framework. Applying the aforementioned concept of *homeostasis* implies tuning a particular set of ODEs to a stable fixed point, which in our case would have been particularly hard on arbitrarily large sizes. Moreover, we could have not taken advantage of the networks applications that exists in the literature mentioned above either, for two main reasons:

- These networks are an abstract representation of the coupling of terms in a system of ODEs, whereas, in our case a network represents physical connections, with a real weight associated to each connection (e.g. synapses of a neuron with their the firing rate).
- These networks are usually limited to small sizes. On the other hand, we speak of arbitrarily large networks, possibly with diverging number of vertices. This is why, to validate our predictions, we have run numerical simulations on both real and synthetic networks of $O(10^4)$ nodes, but the theory is not size limited.

We remark that we are using the term as “homeostatic plasticity”, a notion used in biology that specifically refers to the capacity of neurons to regulate their own activity (G. Turrigiano (et al.), ”Homeostatic plasticity in the developing nervous system” <https://www.nature.com/articles/nrn1327>). This has been clarified in the Introduction section.

Overall, we think that the overlap between our model and the concept of homeostasis used in the literature mentioned above is very limited. However, we also believe that using the word homeostasis can be misleading for the reader. For this reason we have replaced the word *homeostasis* with the more specific phrasing *homeostatic plasticity*.

According to this modification we changed the current title to *A theory for percolation in networks with local homeostatic plasticity*. We also added a clarification in the Introduction section in order to avoid any misinterpretation by the reader.

4. Lack of a significant implication. Because the problem considered is too simple to be generalized and too abstract to relate to real-world systems, the implications that follow from it are also very limited.

As we already explained in point 2, with this paper we want to present a theoretical contribution to the understanding of the implications of damage-response processes, just as percolation theory is a cornerstone for understanding the resilience (to damage only) of complex networks, even if every real-life scenario will very likely have some additional flavour to the abstract picture that percolation theory presents.

My opinion would be different if the paper validated the model against at least one real-world network, or if it established sufficiently general results on network adaptation to damage (not limited to thresholding) that could constitute a new addition to network theory.

This is indeed a very delicate point, thanks for pointing it out. Our work is meant to shed

light, at a theoretical level, to adaptive degradation processes that happen in real networks, but that are impossible to reproduce experimentally at a laboratory level. This is indeed a common feature of theoretical contributions of this kind. For instance in the famous paper “Error and attack tolerance of complex networks”, by Réka Albert, Hawoong Jeong & Albert-LászlóBarabási (<https://www.nature.com/articles/35019019>), the authors describe a situation very similar to ours, in which a network is subjected to random failure or targeted attacks. To validate their theoretical predictions, attacks or failures were virtually imposed on both real and synthetic networks, since doing large enough experiments to ensure significant statistics is not feasible. This is exactly the same guideline that we followed. At very small scales we know that synaptic scaling has been observed and documented extensively (Teller, Sara (et. al), “Spontaneous Functional Recovery after Focal Damage in Neural Cultures” <https://www.eneuro.org/content/7/1/ENEURO.0254-19.2019>, Keck, Tara (et. al) “Synaptic Scaling and Homeostatic Plasticity in the Mouse Visual Cortex In Vivo”, <https://pubmed.ncbi.nlm.nih.gov/24139037/>), therefore it is important to ask ourselves how local adaptation rules of this kind translate at arbitrarily large sizes, which is exactly the theoretical gap we are trying to fill. We added a clarification on this point on the Discussion section.

We hope that after providing additional information and clarifications, the Referee will reconsider their opinion. Thanks for the work on revising our manuscript, we much appreciate it.

Sincerely,

Alex Arenas, on behalf of the authors

REVIEWERS' COMMENTS

Reviewer #3 (Remarks to the Author):

I appreciate the authors' effort to explain their work in the context of my comments. I accept the authors' explanation that the simple model considered in this paper is a theoretical abstraction inspired by homeostatic plasticity in neurons. But in neuroscience, homeostatic plasticity is an experimental reality for which data is available. I did not expect the paper to report new experiments, but the fact that this model cannot provide insight into existing data is a weakness. The situation was different for some of the papers cited in the responses (such as the "error and attack" paper) since in those cases the results had a strong connection with real systems.

The authors provided long explanations (which I appreciate), but the paper did not change substantially and consequently neither did my main concerns. Having noted that, I don't strongly object to publishing this paper. However, I do have strong feelings against the current title since it overstates the content of the paper. Instead of

"A theory for percolation in networks with local homeostatic plasticity"

a more accurate title would be

"An example of percolation in networks with local homeostatic plasticity"

Indeed, what the paper considers is really an example (the simple case of thresholding) instead of a general theory. Even if this example includes analytical calculations, that does not elevate it to a theory for percolation.

Referee #3

I appreciate the authors' effort to explain their work in the context of my comments. I accept the authors' explanation that the simple model considered in this paper is a theoretical abstraction inspired by homeostatic plasticity in neurons. But in neuroscience, homeostatic plasticity is an experimental reality for which data is available. I did not expect the paper to report new experiments, but the fact that this model cannot provide insight into existing data is a weakness. The situation was different for some of the papers cited in the responses (such as the "error and attack" paper) since in those cases the results had a strong connection with real systems. The authors provided long explanations (which I appreciate), but the paper did not change substantially and consequently neither did my main concerns. Having noted that, I don't strongly object to publishing this paper. However, I do have strong feelings against the current title since it overstates the content of the paper. Instead of "A theory for percolation in networks with local homeostatic plasticity"

a more accurate title would be

"An example of percolation in networks with local homeostatic plasticity"

Indeed, what the paper considers is really an example (the simple case of thresholding) instead of a general theory. Even if this example includes analytical calculations, that does not elevate it to a theory for percolation.

We thank the referee for her/his throughfull revision of our manuscript, with no doubt these comments have helped to improve the manuscript readability and contents. Following the suggestion, we decided to restrict our title accordingly as: “Percolation in networks with local homeostatic plasticity”

Sincerely,

Alex Arenas, on behalf of the authors